# Investigation and Follow-Up of a Staphylococcal Food Poisoning Outbreak Linked to the Consumption of Traditional Hand-Crafted Alm Cheese

**DOI:** 10.3390/pathogens9121064

**Published:** 2020-12-19

**Authors:** Virginia Filipello, Emanuela Bonometti, Massimo Campagnani, Irene Bertoletti, Angelo Romano, Fabio Zuccon, Chiara Campanella, Marina Nadia Losio, Guido Finazzi

**Affiliations:** 1Istituto Zooprofilattico Sperimentale della Lombardia e dell’Emilia Romagna, Headquarters, 25124 Brescia, Italy; emanuela.bonometti@izsler.it (E.B.); chiaracampanella@libero.it (C.C.); marinanadia.losio@izsler.it (M.N.L.); guido.finazzi@izsler.it (G.F.); 2National Reference Centre for Emerging Risks in Food Safety, Istituto Zooprofilattico Sperimentale della Lombardia e dell’Emilia Romagna, 25124 Brescia, Italy; 3Azienda di Tutela della Salute della Montagna, 22014 Dongo, Italy; m.campagnani@ats-montagna.it; 4Istituto Zooprofilattico Sperimentale della Lombardia e dell’Emilia Romagna, Sondrio Chapter, 23100 Sondrio, Italy; irene.bertoletti@izsler.it; 5National Reference Laboratory for Coagulase Positive Staphylococci, Istituto Zooprofilattico Sperimentale del Piemonte, Liguria, e Valle d’Aosta, 10154 Torino, Italy; angelo.romano@izsto.it (A.R.); fabio.zuccon@izsto.it (F.Z.)

**Keywords:** staphylococcal food poisoning, enterotoxins, multilocus sequence typing, foodborne, diseases, *Staphylococcus aureus*

## Abstract

Staphylococcal food poisoning (SFP) is one of the most important foodborne diseases. This work describes a SFP event linked to the consumption of alm cheese and involved three people belonging to the same family. Leftovers of the consumed cheese, samples from the grocery store and the producing alm were collected and tested for Coagulase positive staphylococci (CPS) enumeration and for the presence of staphylococcal enterotoxins (SEs). Isolates were typed with MLST, spa typing, and tested for SEs and methicillin resistance genes. An in vitro test evaluated SEs production in relation to bacterial growth. The presence of CPS and SEs was detected in all cheese samples and all isolates belonged to the same methicillin sensitive ST8/t13296 strain harbouring *sed*, *ser* and *sej* genes. The in vitro test showed the production of enterotoxins started from 10^5^ CFU/mL. The farmer was prescribed with corrective actions that led to eradication of the contaminating strain.

## 1. Introduction

Staphylococcal food poisoning (SFP) is one of the most important foodborne diseases. It is caused by the ingestion of preformed staphylococcal enterotoxins (SEs), thermostable proteins produced by enterotoxigenic strains of coagulase-positive staphylococci (CPS) mainly *Staphylococcus aureus*. The onset of symptoms occurs within a few hours causing nausea, vomiting and diarrhea, and the disease severity is SEs concentration dependent. Generally, the intoxications are self-limiting within 24 h; however, it might be fatal in children and in the elderly (0.03 to 4.4% of cases) [1]. *S. aureus* is ubiquitous, it can colonize the human and animal skin and mucosae, the udder causing mastitis, and environmental surfaces through biofilm production. To date, 28  SEs and SE-like toxins (SEls) have been reported in literature; however, only the five so-called classical enterotoxins SEA, SEB, SEC, SED, and SEE can be detected using commercial immune-assays [2]. According to the data published by EFSA and ECDC, bacterial toxins were the second cause of foodborne outbreak in 2017, and in Italy show an increasing trend [3]. Historically, SFP has been linked to improper food handling by operators acting as carriers. However, lately SFP was often linked to dairy products, in which the presence of enterotoxigenic *S. aureus* strains agent of mastitis in the herd can cause final product contamination, where SEs are produced when critical bacterial concentration and environmental temperatures are reached during the processing [4,5]. In this regard, Italy was the country with the highest proportion of cheese and milk samples positive for SEs [3]. Raw milk dairy products are widespread in the Alpine area and are processed manually in small mountain dairies and alms in the summer season, when the cows from the valleys are brought to pastures. These traditional cheeses, namely alm cheeses, are products highly sought after by consumers for the evermore widespread food trends that encourage the search for natural foods and authentic flavors. In some situations, however, the lack of adequate hygiene conditions applied in the production process can increase the probability of contamination with *S. aureus* [6]. This work describes the investigation and long-term management of an SFP episode that occurred in the summer of 2018 in the Lombardy Region and involved three people belonging to the same family.

## 2. Materials and Methods

### 2.1. Background and Case Description

On 3 August 2018, the local Public Health Unit was notified by the local hospital of a suspected SFP. On the previous day, a 2-year-old child experiencing abdominal cramps, vomit, and diarrhea. The same symptoms were reported by the mother (39 year old) on the same day and by the grandfather (78 year old) on the previous day. The mother reported symptoms onset after 4 h from the ingestion of a locally bought alm aged cheese, with remission of symptoms after 12 h from onset. The father and other two daughters did not develop any symptoms and denied consumption of the same cheese.

### 2.2. Epidemiological Investigation and Sampling

Following the report received from the local hospital, the Local Health Authority took steps to perform a targeted sampling to investigate the cause of the intoxication. Specifically, first the residue of cheese consumed by the intoxicated patients was available by the family and a sample from the same shape of cheese was retrieved at the grocery store, were collected. It was possible to trace back the producing alm from the grocery store, where samples from the same batch and five additional different batches of the same type of cheese were collected at different stages of ripening, with lot 1 being the most aged and lot 6 the freshest. A sample of raw milk, samples from six environmental production surfaces and from the cheesemaker’s hands were also collected (Table 1).

### 2.3. Laboratory Testing

Coagulase positive staphylococci (CPS) count on RPF Baird Parker plates (ISO 6888-2:1999 Amd 1:2003) was performed on all samples, while detection of the classical SEs by VIDAS^®^ (bioMerieux, Marcy-l’Étoile, France, according to ISO 19020:2017) was performed on all food samples. From the positive plates, a typical isolated colony from each sample was plated on blood agar plates to obtain isolates for further molecular typing. Nucleic acids were extracted by boiling procedure: briefly, a colony from each blood agar plate was dissolved in 100 µL Chelex^®^ (Bio-Rad, Hercules, CA, USA), incubated for 1 h at 56 °C, then for 45 min at 99 °C, and finally spun down for 5 min at 13,000× *g*. The supernatant was collected and stored at 4 °C until use. For each isolate, species identification was confirmed by PCR of the *nuc* gene, then all isolates were tested for the presence of 11 SEs genes (*sea, seb, sec, sed, see, ser, seg, seh, sei, sej, sep*) and of methicillin resistance genes *mecA* and *mecC* [7,8,9]. Moreover, all isolates were subtyped with MLST and *spa* typing according to Enright et al. and Shopsin et al., respectively [10,11]. An in vitro test to evaluate the strain production of enterotoxins as a function of microbial concentration was carried out. Briefly, from a 0.5 McFarland (corresponding to 1.5 × 10^8^ cfu/mL) bacterial solution in BPW, 1:10 serial dilutions were prepared to reach a starting inoculum of about 10^4^ cfu/mL in a 50 mL volume. The prepared solution was then incubated at 37 °C in agitation and sampled at 0, 1, 2, 3, 4, 5, 6, 7, 8, 9, 12, 24, 32 and 48 h after inoculum. Specifically, 1 mL solution was destined to plate counting and 500 µL were used for detection of SEs as previously described.

## 3. Results

The presence of SED was detected in all cheese samples. Results were confirmed by the NRL for CPS (Istituto Zooprofilattico Sperimentale del Piemonte, Liguria e Valle d’Aosta, Turin, Italy) and by the EURL for CPS (Anses, Maisons-Alfort, France). No SE was detected in the raw milk sample. CPS were detected in all food samples with contamination ranging from 10^1^ to 10^4^ cfu/g, the highest concentration of SED was found in the cheese shape consumed by the patients (Table 1).

As for the environmental samples, only from the cheese mold and from the cheesemaker’s hands was it possible to isolate *S. aureus*. All isolates were negative for the presence of methicillin resistance genes, but harbored the *sed*, *ser* and *sej* SEs genes. Subtyping identified all isolates as belonging to ST8/t13296 as defined by MLST and spa typing, respectively. According to the in vitro growth assay, the production of enterotoxin for the strain tested started when the microbial concentration reached 10^5^ cfu/mL (Figure 1).

## 4. Discussion

The investigation led to identify the responsible food and trace back the cheese production site involved in the outbreak. The raw milk sample was contaminated with 10^3^ cfu/mL; this result is consistent with the contamination level of raw milk in alms found in a previous survey [6]. Such level of contamination does not determine SEs production but it allows the rapid onset of exponential growth in the early phases of the cheese-making process. Indeed, the milk is heated on a wood fire to facilitate the curd formation, thus setting the perfect condition to reach the critical contamination level for SEs production (i.e., 10^5^ cfu/mL), as confirmed by the in vitro assay and also reported in literature [5]. In this regard, the production technology of alm cheese is not effective in preventing and limiting the proliferation of *S. aureus* in the early stages of the cheese-making process, but actually increases the risk of bacterial growth. Indeed, alm cheese production is not standardized by a procedural guideline and the process can differ from alm to alm. As an example, milk is generally kept at room temperature overnight to allow cream surfacing, and there is no fixed temperature to be reached when warming the milk to obtain the curd; these are critical steps that might put at risk the safety of the final product. The different lots of cheese had a *S. aureus* contamination that ranged 10^1^–10^4^ cfu/g, depending on the ripening of the shape. Indeed, during the aging process, the bacterial populations, *S. aureus* among them, tend to decrease due to the dehydration. Interestingly, lot 6 cheese had the higher CPS count but SEs were not detected. In fact, this cheese lot has been sampled shortly after production, therefore the natural reduction of bacterial populations that occurs during the ripening process did not yet occur. The critical bacterial concentration for SEs production is 10^5^ cfu/g, as also demonstrated by the in vitro assay. It is, therefore, possible that, for this cheese lot, the lower contamination did not trigger SEs production. Indeed, the EU regulation (i.e., Commission Regulation EC n. 2073/2005, [12]) foresees the detection of SEs in raw milk cheese only in case of *S. aureus* values higher than 10^5^ cfu/g. In our case however, the *S. aureus* count of the cheese sold was about 10^1^ cfu/g and the critical situation at the producing alm would have been ignored had an infant not been admitted to an emergency ward. The farmer was prescribed with corrective actions to enhance hygienic measure on the plant and improve herd management, outlining a rather fortuitous notification of the problem. ST8 identifies a *S. aureus* subtype often involved in mastitis and widespread in the bovine population [13]. The corrective actions prescribed by the local health authority were, therefore, primarily directed at ameliorating the herd management. Consequently, 12 out of 20 cows were identified as *S. aureus* carriers with recurring mastitis and, therefore, substituted. Regarding the cheese production, an accurate sanitation of all the equipment and the improvement of the production processing was prescribed. These aspects were verified through an audit on site in the alm premises the subsequent season and had a positive outcome. Finally, it was requested that self-monitoring analysis on the cheese had to be performed before commercialization. For this purpose, a batch of cheese produced for private domestic consumption was tested and found with a very low contamination of a non-toxigenic CPS strain. Moreover, for the subsequent season, tank milk quality parameters were improved and all the tested batches of cheese produced at the alm were found compliant with the hygienic requirements.

## 5. Conclusions

The application of basic good processing practices, starting from the health control of the herd and milking hygiene combined with the surveillance on the product batches, are fundamental to prevent situations that may lead to food poisoning. The application of molecular typing techniques allows a characterization of the isolates by providing important information on the epidemiology of the cases. In the reported case, a condition with a high-risk potential was apparently resolved in an event with moderate impact. This was achieved also thanks to a tight collaboration between authorities (i.e., health care, veterinary services, and official laboratories) joined to a cooperative attitude sported by the private actors, even if it required a great effort for such a small enterprise. It would be desirable to divulge the lessons learnt to improve the overall safety of traditional products, which provide sustenance for the local communities and represent a heritage, to safeguard to the entire population.

## Figures and Tables

**Figure 1 pathogens-09-01064-f001:**
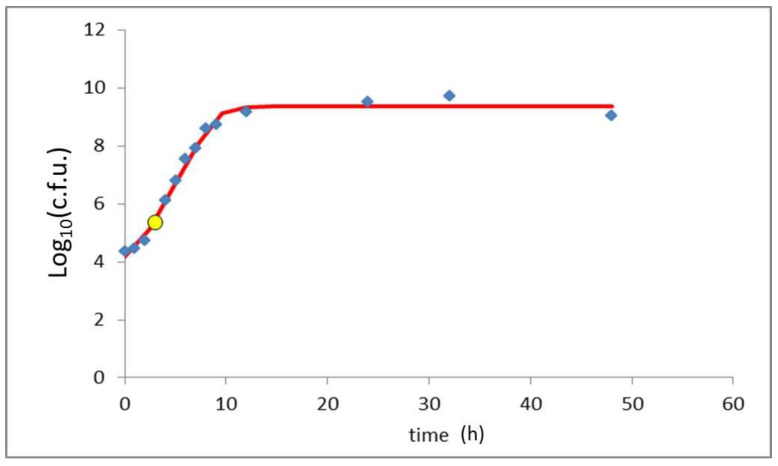
Growth curve of the ST8/t13296 *S. aureus* strain that caused the outbreak. The yellow dot represents the point at which toxinogenesis of SED began.

**Table 1 pathogens-09-01064-t001:** List of the samples collected during the outbreak investigation and analytical results.

ID	Sample	CPS (CFU/g or mL)	SEs Detection	SEs Quantification ^1^
220443	Cheese retrieved at household (lot 1)	30	D	NA
220442	Cheese from deli store (lot 1)	NA	D	2.978
233694	Cheese from alm (lot 1)	240	D	1.820
246679	Cheese from alm (lot 2)	570	D	0.047
246681	Cheese from alm (lot 3)	1300	D	2.292
260114	Cheese from alm (lot 4)	3100	D	1.288
260145	Cheese from alm (lot 5)	1400	D	1.571
260160	Cheese from alm (lot 6)	55,000	D	<LoQ
254209	Raw milk from alm	2300	-	NA
254211	Cheesemaker hands	+	NA	NA
260173-1	Cheese vat	-	NA	NA
260173-2	Cheese mold	+	NA	NA
260173-3	Drainer	-	NA	NA
260173-4	Bucket	-	NA	NA
260173-5	Mixing stick	-	NA	NA
260173-6	Curd paddle	-	NA	NA

^1^ As provided by the EURL for CPS (Anses, Maisons-Alfort, France); LOD 0.008 ng/g, LOQ 0.028 ng/g.

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
