# Peer review of "Investigation and Follow-Up of a Staphylococcal Food Poisoning Outbreak Linked to the Consumption of Traditional Hand-Crafted Alm Cheese"

_pathogens, 2020, doi:10.3390/pathogens9121064_

Round 1
Reviewer 1 Report
This paper is very important, well-designed and well-presented.
However, it could be greatly improved (for the non-specialist) and be more transparent, if the authors would also add the respective references in the methods section.
If possible, the quantitative specifications could be improved by any information on sensitivity, e.g. cut-off range, or any indication from a statistical point (SD, etc.).
Author Response
Dear reviewer, thanks for the time spent in ameliorating our work. Please find following the reply to your comments.
However, it could be greatly improved (for the non-specialist) and be more transparent, if the authors would also add the respective references in the methods section.
Thanks for the observation, the references have been added to the methods section (see lines 82 and 83)
If possible, the quantitative specifications could be improved by any information on sensitivity, e.g. cut-off range, or any indication from a statistical point (SD, etc.).
For the CPS count the analysis is carried out with an ISO procedure and since are routine samples they are tested singlularly, so no statistic information are available. The LOD and LOQ of the SE quantification has been instead added in the footnotes of Table 1 (see line 97).
Reviewer 2 Report
The authors present a simple but well described paper identifying a CPS producing enterotoxin outbreak.
Minor:
- Many symbols (mu, degree, etc.) did not appear.
- Line 82 - missing methods and references
- Table 1 - 'cheese mould' not 'cheese mold'
Other:
- For non-europeans, explaining what alm cheese is might be helpful.
- Cheese lot 6 had ~20-times higher CPS than any other sample but the SE was below the LoQ. Do the authors wish to comment?
Author Response
Dear reviewer, thank for the time spent in ameliorating our work. Please find following detailed answers to your specific comments.
Minor:
- Many symbols (mu, degree, etc.) did not appear.
I think some problems occurred during the uploading process, they have been however corrected again.
- Line 82 - missing methods and references
Sorry for the inattention, the methods have been added in the text and in the reference section
- Table 1 - 'cheese mould' not 'cheese mold'
The spelling has been corrected
Other:
- For non-europeans, explaining what alm cheese is might be helpful.
Thanks for the suggestion, an explanation was already given in the introduction, however, for clarity it has been made explicit (see line 49)
- Cheese lot 6 had ~20-times higher CPS than any other sample but the SE was below the LoQ. Do the authors wish to comment?
This cheese lot has been sampled shortly after production, therefore the natural reduction of bacterial populations that occurs during the ripening process did not yet occur. The critical bacterial concentration for SEs production is 105 cfu/g, as also demonstrated by the in vitro test. It is therefore possible that for this cheese lot the lower contamination did not trigger SEs production. These comments have been added to the discussion (see lines 119-123).
Reviewer 3 Report
General comments: the diagnosis approach and measures adapted in this case report seems to be adequate and the case well described. This paper represents a “communication of an incident for the scientific community” as well for an educational purpose such as mentioned in the conclusions. I suggest some alterations or explanations improving the final merit of this report.
Specific comments
L 30. You are sure? SFP is one important food poisoning, but if you read ESFA reports (see you reference [3], this is not one of the most frequent. Additionally, In subsection 2.5 of https://dx.doi.org/10.3390%2Ftoxins2071751 you can find more information about SFP epidemiology, before 2010 (probably with poor sanitation, hygiene, and poor good practices in installations). Also, this sentence disagrees with the information reported in L38-39. So, I suggest using the term “important” instead of “frequent” in L30 as well in L15 (abstract).
L 69- The raw milk is normally stored using refrigeration (at ⁓4 Celsius)? This information is important regarding the level of 103 cfu/mL(L 108), and indirectly the information of L127.
L114-118: You are sure that the “production technology of alm cheese is not effective in preventing and limiting the proliferation of S. aureus in the early stages of the cheese-making process, but actually increases the risk of bacterial growth.” I suggest adding a reference for this.
L116-117: “The different lots of cheese had S. aureus 117 contamination that ranged 101 - 104 cfu/g, depending on the ripening of the shape”. Where is this information in M&M? table 1 reports CPS.
L121 You mean 104 cfu/g? Please see the previous comment
Author Response
Dear reviewer, thanks for the time spent in ameliorating our work. Please find following detailed answers to your comments
- L 30. You are sure? SFP is one important food poisoning, but if you read ESFA reports (see you reference [3], this is not one of the most frequent. Additionally, In subsection 2.5 of https://dx.doi.org/10.3390%2Ftoxins2071751 you can find more information about SFP epidemiology, before 2010 (probably with poor sanitation, hygiene, and poor good practices in installations). Also, this sentence disagrees with the information reported in L38-39. So, I suggest using the term “important” instead of “frequent” in L30 as well in L15 (abstract).
Thanks for the observation and suggestion, the term has been substituted.
- L 69- The raw milk is normally stored using refrigeration (at ⁓4 Celsius)? This information is important regarding the level of 103 cfu/mL(L 108), and indirectly the information of L127.
No, actually for alm cheese the milk is kept at room temperature overnight to allow cream surfacing, and then is processed for cheese production. This information has been added to the discussion following your next comment (see lines 117-121)
- L114-118: You are sure that the “production technology of alm cheese is not effective in preventing and limiting the proliferation of S. aureus in the early stages of the cheese-making process, but actually increases the risk of bacterial growth.” I suggest adding a reference for this.
Alm cheese production is not standardized by a procedural guideline. Unfortunately there are no references to cite that describe the process, it is indeed different from alm to alm. As an example, there is no fixed temperature to be reached when warming the milk to obtain the curd. This is of course a critical step that might jeopardize the safety of the final product. These comments have however been added to the discussion section (see lines 117-121).
- L116-117: “The different lots of cheese had S. aureus 117 contamination that ranged 101 - 104 cfu/g, depending on the ripening of the shape”. Where is this information in M&M? table 1 reports CPS.
The CPS count method is reported in line 73. For clarity an indication on the ripening has been added in the methods section in lines 70 and 71.
- L121 You mean 104 cfu/g? Please see the previous comment
No, the cheese sold had a contamination of 101 cfu/g, the typo has been corrected (see line 131)